# Species-Level Taxonomic Characterization of Uncultured Core Gut Microbiota of Plateau Pika

Ji Pu,[a] Jing Yang,[a,b] Shan Lu,[a,b] ⬤Dong Jin,[a,b] Xuelian Luo,[a] ⬤Yanwen Xiong,[a] ⬤Xiangning Bai,[a] Wentao Zhu,[a] Yuyuan Huang,[a] Shusheng Wu,[c] Lina Niu,[d] ⬤Liyun Liu,[a] ⬤Jianguo Xu[a,b,e]

[a]State Key Laboratory of Infectious Disease Prevention and Control and National Institute for Communicable Disease Control and Prevention, Chinese Center for Disease Control and Prevention, Beijing, China
[b]Research Units of Discovery of Unknown Bacteria and Function, Chinese Academy of Medical Sciences, Beijing, China
[c]Yushu Prefecture Center for Disease Control and Prevention, Yushu, China
[d]Key Laboratory of Tropical Translational Medicine of Ministry of Education, Hainan Medical University, Haikou, China
[e]Institute of Public Health, Nankai University, Tianjing, China

**ABSTRACT** Rarely has the vast diversity of bacteria on Earth been profiled, particularly on inaccessible plateaus. These uncultured microbes, which are also known as "microbial dark matter," may play crucial roles in maintaining the ecosystem and are linked to human health, regarding pathogenicity and prebioticity. The plateau pika (*Ochotona curzoniae*) is a small burrowing steppe lagomorph that is endemic to the Qinghai-Tibetan Plateau and is a keystone species in the maintenance of ecological balance. We used a combination of full-length 16S rRNA amplicon sequencing, shotgun metagenomics, and metabolomics to elucidate the species-level community structure and the metabolic potential of the gut microbiota of the plateau pika. Using a full-length 16S rRNA metataxonomic approach, we clustered 618 (166 ± 35 per sample) operational phylogenetic units (OPUs) from 105 plateau pika samples and assigned them to 215 known species, 226 potentially new species, and 177 higher hierarchical taxa. Notably, 39 abundant OPUs (over 60% total relative abundance) are found in over 90% of the samples, thereby representing a "core microbiota." They are all classified as novel microbial lineages, from the class to the species level. Using metagenomic reads, we independently assembled and binned 109 high-quality, species-level genome bins (SGBs). Then, a precise taxonomic assignment was performed to clarify the phylogenetic consistency of the SGBs and the 16S rRNA amplicons. Thus, the majority of the core microbes possess their genomes. SGBs belonging to the genus *Treponema*, the families *Muribaculaceae*, *Lachnospiraceae*, and *Oscillospiraceae*, and the order *Eubacteriales* are abundant in the metagenomic samples. In addition, multiple CAZymes are detected in these SGBs, indicating their efficient utilization of plant biomass. As the most widely connected metabolite with the core microbiota, tryptophan may relate to host environmental adaptation. Our investigation allows for a greater comprehension of the composition and functional capacity of the gut microbiota of the plateau pika.

**IMPORTANCE** The great majority of microbial species remain uncultured, severely limiting their taxonomic characterization and biological understanding. The plateau pika (*Ochotona curzoniae*) is a small burrowing steppe lagomorph that is endemic to the Qinghai-Tibetan Plateau and is considered to be the keystone species in the maintenance of ecological stability. We comprehensively investigated the gut microbiota of the plateau pika via a multiomics endeavor. Combining full-length 16S rRNA metataxonomics, shotgun metagenomics, and metabolomics, we elucidated the species-level taxonomic assignment of the core uncultured intestinal microbiota of the plateau pika and revealed their correlation to host nutritional metabolism and adaptation. Our findings provide insights into the microbial diversity and biological significance of alpine animals.

Address correspondence to Jianguo Xu, xujianguo@icdc.cn.

The authors declare no conflict of interest.

**KEYWORDS** 16S rRNA, *Ochotona curzoniae*, species-level, metabolomics, metagenomics, metataxonomics, microbiota, plateau pika, uncultured

The great majority of microbial species in the human and animal microbiota remain uncultured, severely limiting their taxonomic characterization as well as their biological and medical understanding (1). The current classification has been limited to cultured taxa, which might only represent a small proportion of the microbiota that are prone to grow under the applied cultivation conditions.

The composition of the microbial community is usually elucidated via high-throughput, short-reads 16S rRNA sequencing, in which the amplicons of 16S rRNA fragments contain one to three hypervariable regions. The partial 16S rRNA does not carry enough evolutionary information to resolve taxonomic ranks to the species-level, and the databases that are currently available lack high-quality reference sequences for uncultured taxa in specific ecosystems, thereby hampering comprehensive taxonomic assignment (2). A metataxonomic approach that combines full-length 16S rRNA gene amplicon sequencing along the PacBio platform with the application of the operational phylogenetic unit (OPU) procedure has been developed to taxonomically characterize the complex microbiota, especially for uncultured bacterial taxa. Our previous studies, which employed this metataxonomic approach, have primarily unveiled numerous uncultured gut bacteria of wild animals in the Qinghai-Tibetan Plateau (3, 4).

Although full-length 16S rRNA could provide a species-level structure of the uncultured microbiota without cultivation (5), it still lacks sufficient information with which to fully appreciate these essential organisms' roles. With the progress of high-throughput sequencing technology, we now have the opportunity to study the entire DNA gallery of given samples. By using the state-of-the-art metagenomic toolbox, we can bin the *de novo* assembled contigs into highly complete species-level genome bins (SGBs) that are comparable in quality to the genomes of cultured organisms (6–8). By coupling ANI and AAI with the phylogeny approaches, we can robustly assess taxonomic rank and adequately place the SGBs in the tree of life. Through these efforts, the stable and reliable taxonomic assignment of uncultured bacterial species can be brought into reality (9, 10).

The plateau pika (*Ochotona curzoniae*), also called the "black-lipped pika", is a small, diurnal, nonhibernating, burrowing steppe lagomorph that is highly distributed in the alpine meadow of the Qinghai-Tibetan Plateau (11). Considered a keystone species, plateau pika is crucial for the maintenance of the stability of the alpine ecosystem (12). The microbiota inhabiting the gut of the plateau pika may play a central role in the host nutritional metabolism and adaptation (Fig. S3). Studying the plateau pika's intestinal flora is crucial to exploring the microbial diversity and biological significance of wild alpine animals.

In this study, we conducted a comprehensive investigation of the gut microbiota of the plateau pika via a multiomics endeavor. We established a high-quality, species-level, full-length 16S rRNA and SGBs data set that provides novel insights into the diversity and functions of the core uncultured gut microbiota of the plateau pika.

## RESULTS

**Species-level assignments of plateau pika gut microbiota.** A total of 105 intestinal content samples of plateau pika were sequenced. The PacBio RSII platform generated 1,408,058 raw, full-length 16S rRNA gene reads. After quality trimming and chimera removal, 1,035,221 (73.52%) high-quality reads (error rate of 0.15%) (Fig. S1) were acquired, with an average of 9,326.32 $\pm$ 5,214.25 reads per sample. The average length was 1,445 $\pm$ 4.83 bp (Table S1). The high-quality reads were clustered into 23,277 OTUs, with a 98.7% similarity cutoff. The representative sequences of each OTU were picked out for phylogenetic inference. Via the manual inspection of the *de novo* tree, we identified 618 OPUs, with a mean of 166 OPUs $\pm$ 35 per sample (Table S2).

Taxonomically, we divided the 618 OPUs into three categories (Fig. 1A–C). (i)

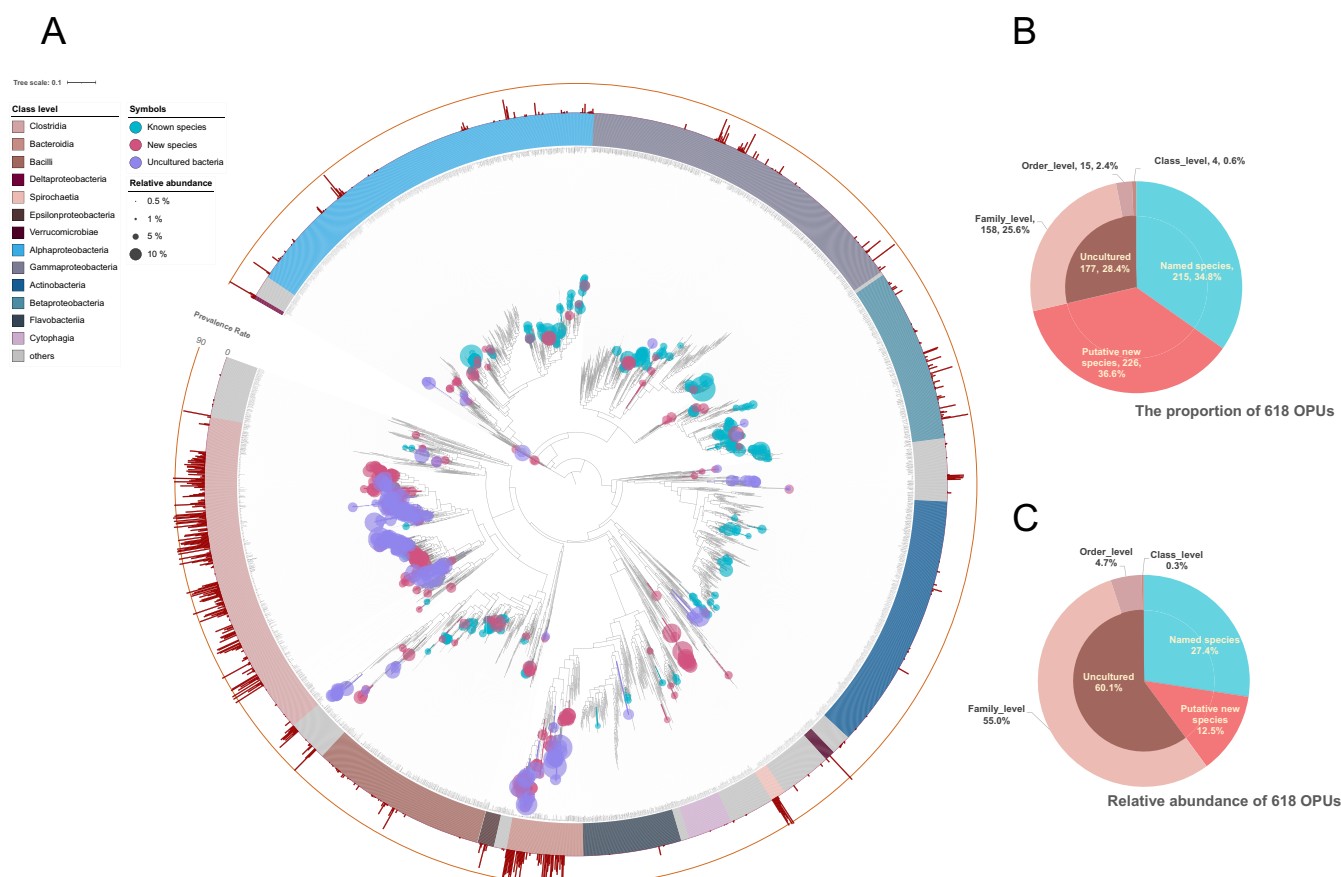

**FIG 1** Taxonomic profile of plateau pika gut microbiota. (A) Full-length 16S rRNA gene phylogenetic trees of 215 known species, 226 potentially new species, and 177 uncultured bacteria. (B) The proportion of OPU numbers. The numbers of OPUs classified into Named, Putative new, and Uncultured bacteria were 215, 226, and 177, respectively. (C) The relative abundance of OPUs. The relative abundance values of the OPUs classified into Named, Putative new, and Uncultured bacteria were 27.4%, 12.5%, and 60.1%, respectively. Each dot represents the relative abundance of a unique OPU. Three categories of OPUs, stained with different colors.

Known species (215 OPUs; relative abundance of 27.43%), affiliated with the type strain of known bacterial species (>98.7% identity), belonging to 115 genera, 62 families, 27 orders, 12 classes, and 5 phyla. These accounted for generally low relative abundance values (0.07%), on average. (ii) Potentially new species (226 OPUs; relative abundance of 12.47%), embedded in known genera that formed a monophyletic lineage, belonging to 107 genera, 68 families, 38 orders, 24 classes, and 14 phyla. (iii) Uncultured bacteria (176 OPUs; relative abundance of 60.10%) that consisted of sequences that were not affiliated with any known genus and formed distinct lineages that would represent new genera, families, or higher taxa. The 176 uncultured bacteria comprised 158 OPUs at the family level (relative abundance of 55.06%), 15 OPUs at the order level (relative abundance of 4.75%), and 4 OPUs at the class level (relative abundance of 0.30%). Taken together, the 618 OPUs were affiliated with 216 genera, 112 families, 56 orders, 33 classes, and 16 phyla. The uncultured bacteria and potentially new species occupied an abundance of over 80% of the entire microbial community. These to-be-revealed microorganisms predominated in the gut of the plateau pika.

**The core taxa composition.** Of the 618 OPUs, 39 OPUs were prevalent in more than 90% of samples, with these OPUs accounting for over 60% (66.58 ± 23.97%) of the relative abundance of bacterial taxa found in the samples (Fig. 1D, 1E, and 4A). Notably, the core microbiota with 39 OPUs had none of the known bacterial species (Fig. S4; Table S3). They were either potentially new species or uncultured higher taxa. These novel bacteria affiliated with strict anaerobic lineages constituted the main structure of the pika intestinal flora. They assigned to the uncultured order at *Mollicutes* (relative abundance of 0.23%);

the uncultured family at *Eubacteriales* (relative abundance of 4.79%) and *Bacteroidales* (relative abundance of 12.88%); the uncultured genera at *Oscillospiraceae* (relative abundance of 12.10%), *Lachnospiraceae* (relative abundance of 9.36%), *Muribaculaceae* (relative abundance of 8.63%), *Prevotellaceae* (relative abundance of 6.47%), *Christensenellaceae* (relative abundance of 5.47%), *Akkermansiaceae* (relative abundance of 1.85%), and *Desulfovibrionaceae* (relative abundance of 0.31%); and the new species at *Treponema* (relative abundance of 3.99%), *Alistipes* (relative abundance of 0.20%), and *Eubacterium* (relative abundance of 0.28%) (Table S1). Hence, we intriguingly intend to acquire the genomes of these highly distributed and uncultured species for deep analysis.

**Recovering SGBs from the plateau pika gut microbiome.** A mean of 145,938 $\pm$ 35,625 contigs ($\geq$1 kb) from each metagenomic set were independently assembled with an average of 46.96 $\pm$ 6.49 million quality-filtered reads (paired-end, 150 bp) per sample (sample size of 16) (Table S4). By integrating three different algorithms implemented in the Metawrap toolkit, we recovered and reassembled 766 MAGs from the assembled contigs. After dereplication with a 99% average nucleotide identity (ANI) threshold and a standardized genome quality (>50% genome completeness, <10% contamination and completeness, 5 · contamination $\geq$50) (13), the represented 527 nonredundant genomes were selected as the qualified draft genomes. To further determine how many species were contained in these 16 metagenomic collections, we clustered all 527 of the genomes into species-level genome bins (SGBs) via the dRep strategy, with an ANI threshold of 95% differentiating the species boundary. The clustering procedure resulted in a total of 360 inferred microbial species, and only the nearly complete 109 high-quality (>90% genome completeness, <50% contamination) SGBs were optimized for downstream analyses. Their quality information is given in Fig. S2 and Table S5.

Combining genome-based taxonomy (GTDB-Tk), ANI calculations, AAI calculations, and phylogenomic analyses (Fig. S6; Table S7), we conducted precise taxonomic annotation for these 109 high-quality SGBs and assigned each of them a name. Overall, the 109 SGBs affiliated with 5 bacterial phyla (*Bacillota*, 62; *Bacteroidota*, 36; *Spirochaetota*, 6; *Pseudomonadota*, 4; *Verrucomicrobiota*, 1) (Fig. 2) and 20 out of the 109 (18.35%) SGBs (Fig. 3) represented the same dominant core microbial lineages as were previously determined by the full-length 16S rRNA gene amplicon sequencing (Fig. 4; Fig. S5). Of the 109 SGBs, 32 were classified as new species (ANI and AAI both <95%), 57 were classified to the family level (AAI < 65%), and 20 were classified to the order level (AAI < 50%). They were assigned to the uncultured family at *Eubacteriales* (relative abundance of 3.64%); the uncultured genera at *Muribaculaceae* (relative abundance of 6.81%), *Lachnospiraceae* (relative abundance of 4.16%), *Oscillospiraceae* (relative abundance of 2.97%), *Prevotellaceae* (relative abundance of 1.37%), and *Lentimicrobiaceae* (relative abundance of 1.13%); and the new species at *Treponema* (relative abundance of 9.48%), *Prevotella* (relative abundance of 2.23%), *Acetatifactor* (relative abundance of 1.98%), and *Alistipes* (relative abundance of 1.01%) (Table S6). Overall, the phylogenomic trees revealed that all of the SGBs were unidentified species, as evidenced by their formation of monophyletic clades, and they achieved significantly lower ANI levels (ranging from 63.41% to 84.48%) and AAI values (ranging from 47.64% to 85.81%) than those of publicly available representative genomes of known species. Such observations underline the phylogenetic novelty of the microbial populations from the intestine of the plateau pika.

**Functional capacity of the plateau pika gut microbiota.** Carbohydrate active enzymes (CAZymes) harvest energy from fiber-rich lignocellulose, as they can assemble, modify, and break down oligosaccharides and polysaccharides. The annotations for carbohydrate-active enzymes (CAZymes) were adopted against the CAZy database. A total of 3,692 glycoside hydrolases (GHs) (32.91 $\pm$ 19.97 per SGB, average identity of 58.48 $\pm$ 11.84%), 1,604 glycosyltransferases (GTs) (10.79 $\pm$ 5.91 per SGB, average identity of 63.55 $\pm$ 12.08%), 172 polysaccharide lyases (PLs) (1.65 $\pm$ 3.74 per SGB, average identity of 51.75 $\pm$ 9.68%), 469 carbohydrate esterases (CEs) (2.37 $\pm$ 3.14 per SGB, average identity of 62.30 $\pm$ 11.75%), and 211 carbohydrate-binding modules (CBMs) (5.19 $\pm$ 2.94 per SGB, average identity of 54.97 $\pm$ 12.27%) were detected in the plateau

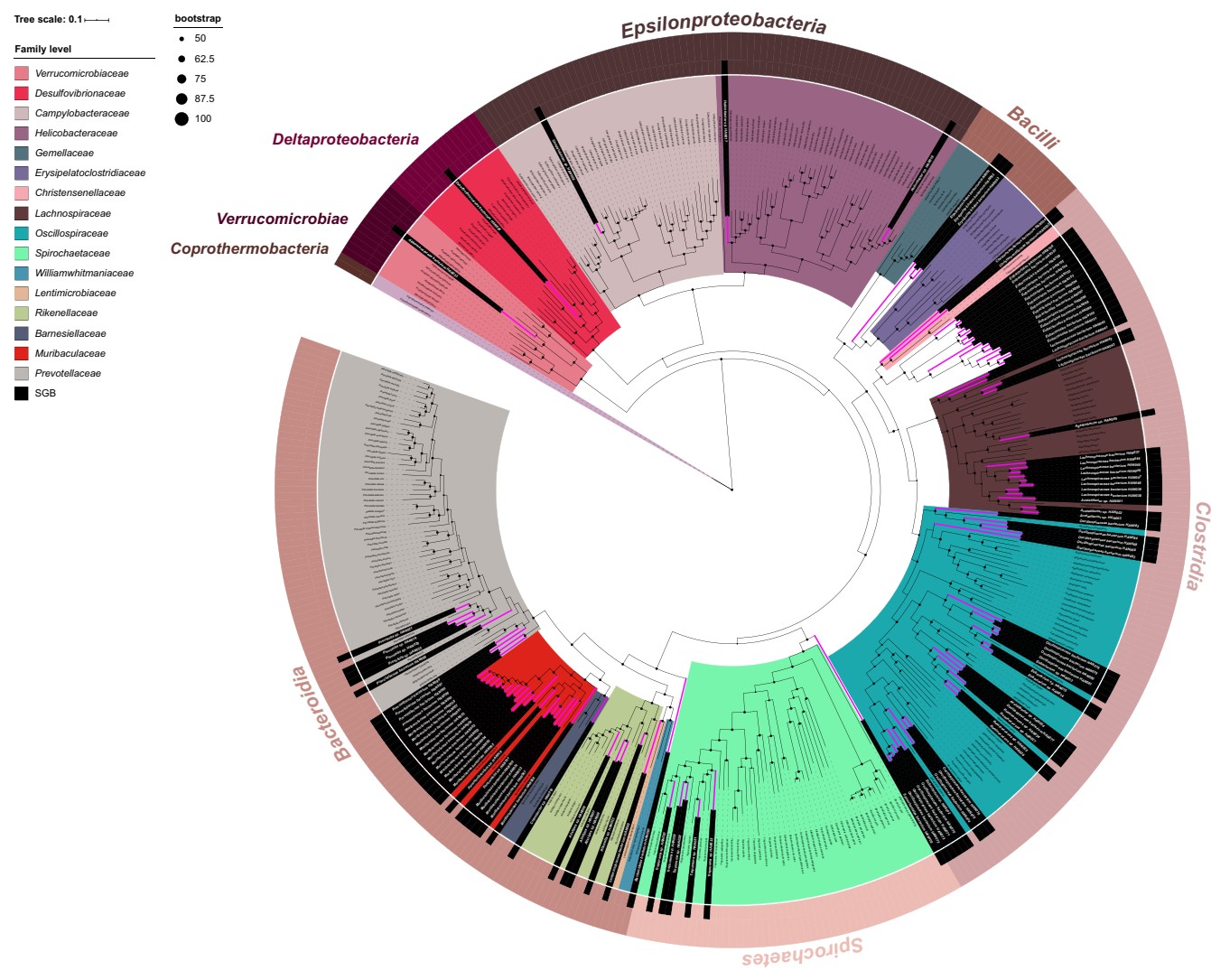

**FIG 2** Phylogeny position of high-quality SGBs. A phylogenomic inference of the 109 high-quality SGBs was conducted from concatenated universal gene sequences using Phylophlan and subsequently drawn using iTOL. The labels colored in black are high-quality SGBs that were retrieved from metagenomic samples, and they additionally incorporate representative genomes of genus type species of 7 classes from the NCBI GenBank.

pika gastrointestinal SGBs (Table S9). The results suggest the existence of a great diversity of CAZymes in the gut microbiota (Fig. S7).

A LC-MS untargeted metabolomics analysis revealed over 1,547 features in intestinal samples from the plateau pika. 33 molecules significantly correlated with SGBs (cooccurrences log conditional probabilities > 4.5). Among them, tryptophan (Fig. 5A) was the most widely connected metabolite with these microbes, and its derivate indoleacrylic acid (IA) was also related to a number of microbes. Therefore, we screened for tryptophan-associated pathways and discovered that 67 out of 109 SGBs contained *trp* operon-related genes and that 38 of them carried the entire gene operon (14) (Fig. 5B). We also detected 6 SGBs carrying the gene operon of the tryptophan reductive fermentation pathway, which consists of a promiscuous set of enzymes containing dehydratases that metabolize tryptophan into IA (15).

## DISCUSSION

In the present study, we profiled the specialized microbial composition in the gastrointestinal tract of the keystone species of the Qinghai-Tibetan Plateau, namely, the plateau pika. Our study involved 105 pika samples and employed a nearly full-length 16S rRNA metataxonomic analysis to explore the precise "species-level" taxonomy. We

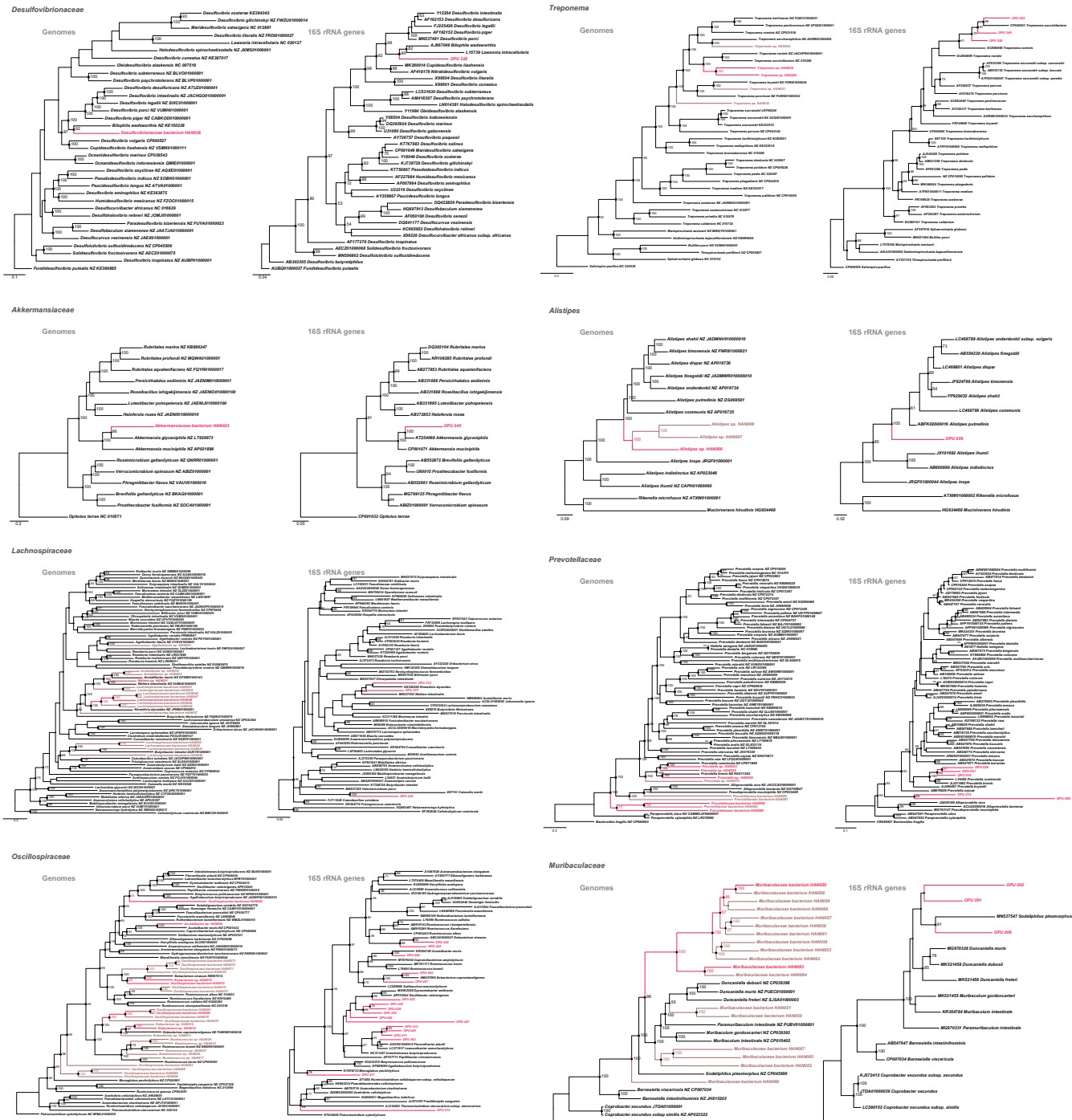

**FIG 3** Phylogenetic trees of core bacteria. The phylogenetic trees are based on core genomes and full-length 16S rRNA gene sequences. The trees showed the precise position of SGBs and core OPUs, compared with members of the most closely related reference genomes. Red labels are core OPUs and corresponding SGBs, whereas known bacteria are shown in black. The phylogenetic trees of the SGBs were generated from core-genome sequences using roary and were inferred using RAxML methods.

preliminarily identified a "core" gut species for the plateau pika, which predominantly consisted of uncultured bacteria. With the help of a shotgun metagenomic strategy, we acquired the SGBs of these core bacteria.

The Qinghai-Tibet Plateau is the highest plateau globally, being 4,500 m above sea level (ASL) on average, and covering about a quarter of the land area of China (2.5 million km²). It is a harsh environment with minimum precipitation, a low temperature

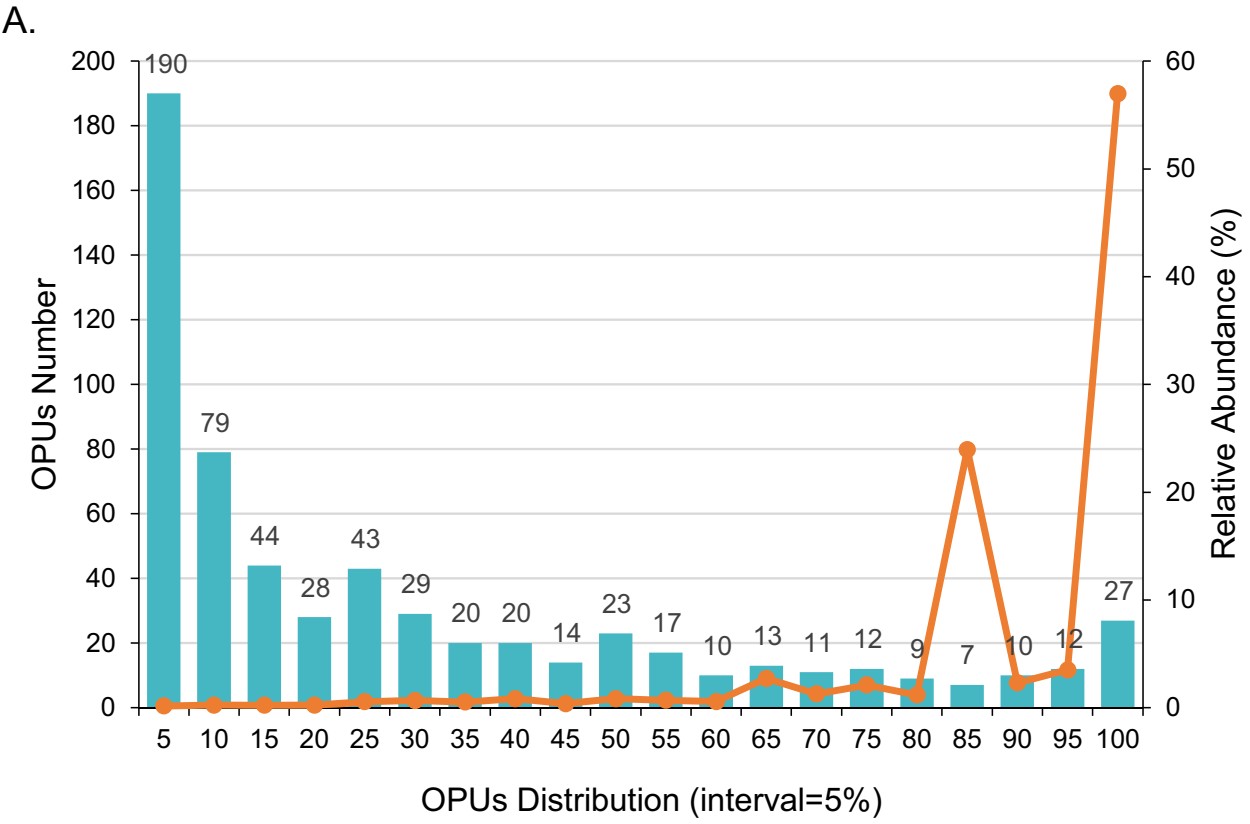

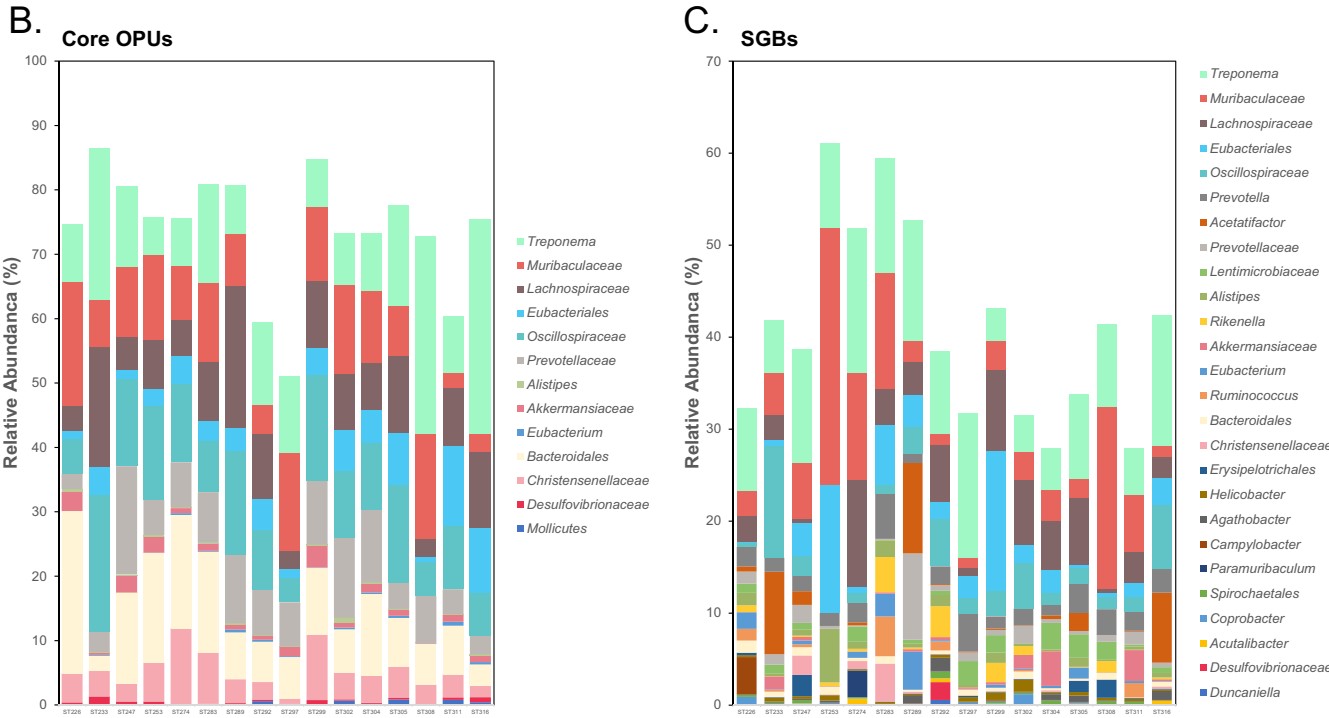

**FIG 4** The distribution of OPUs and SGBs. (A) The numbers (left axis) and the relative abundance (right axis) of the 618 OPUs in each sample, with the 5% interval. Stacked bar plot of the relative abundance of (B) 39 core OPUs and (C) SGBs.

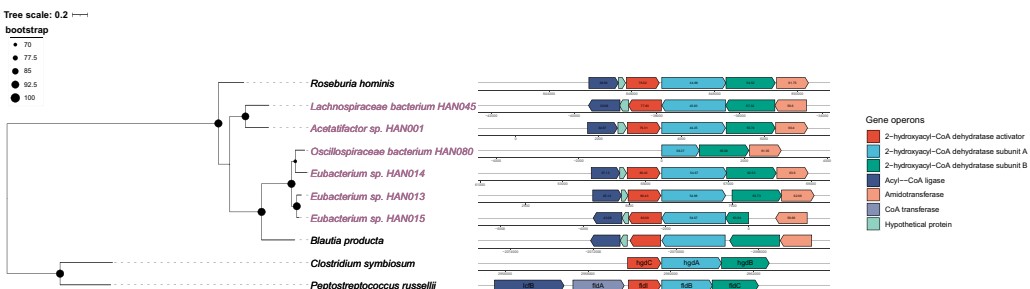

**FIG 5** The MMVEC network and tryptophan metabolism. (A) MMVEC network. The nodes represent the 109 high-quality SGBs, whose abundance is proportional to the size of the node, with a different color representing each family. The squares represent the

that typically falls under −30°C, and an inclement, hypobaric hypoxia atmosphere (16). Nevertheless, plateau pika conquered these complicated physical challenges and were endemic to the alpine meadow, playing vital ecological roles. Under such environmental stress, the gut microbiota that coevolved with the plateau pika may contribute to adaptations related to host metabolism and nutrient harvest (12, 17).

39 OPUs are prevalent across over 90% of plateau pika intestinal samples, and they represent a species-level "core microbiota" (Fig. 4A; Fig. S5A; Table S3). As the plateau pika is a hindgut fermenter, the microbes in the digestive tract of the plateau pika are expected to mediate the degradation and fermentation of high-fiber plants. The metataxonomic assignment of some of these OPUs pointed out that they have a potential ability to digest plant polysaccharides effectively. For instance, members of the families *Oscillospiraceae* and *Prevotellaceae* are ubiquitous in hindgut environments and are well-known for their fiber-degrading capabilities. These taxa were also prevalent in the digestive tracts of American pikas (18). *Treponema* is an abundant genus in the guts of termites, and they are remarkably efficient at degrading recalcitrant woody biomass (19).

An analysis of the CAZymes revealed that the gut microbiota of the plateau pika gut are versatile with complex carbohydrate degradation. The dominant GHs families include starch degrading enzymes (GH13, GH77, and GH97), pectin depolymerizing hydrolases (GH28), pectin, arabinoxylan degradation enzymes (GH43 and GH51), and family GH5, which is primarily composed of cellulases and mannanases. Enzymes from the abundant families GH2 and GH3 function as accessory enzymes to complete plant polysaccharide digestion, and they are involved in the degradation of carbohydrates that originate from microbes in the gut (20). The plateau pika gut microbiota share relatively low similarities (mean of 59.13 ± 12.20%) with the known CAZymes, suggesting that they might be a rich source of novel lignocellulose-degrading enzymes (Fig. S7) (21). The SGBs that are affiliated with the class *Bacteroidia* harbored the highest numbers of CAZymes (70.00 ± 29.65, on average), and these were followed by *Clostridia* (48.98 ± 25.43), *Verrucomicrobiae* (38.00 ± 0.00), and *Spirochaetota* (34.83 ± 9.28) (Fig. S7). Among the class *Bacteroidia*, four SGBs that were annotated to potential new species in *Prevotella* encountered high-level GHs (76.75 ± 25.59, on average) and 47 versatile GHs families, with GH43, GH2, and GH28 being the most abundant. Lineages of the *Prevotella* genus contained common member species of the human and nonhuman gut microbiota, and they have been profoundly implicated in health and disease (22). *Prevotella copri* is among the essential members that are prevalent in over 40% of the human population, and it is associated with high-fiber diets (5, 23). While in the plateau pika, the SGBs of *Prevotella* accounted for a 2.23% relative abundance, and they were affiliated with *P. ruminicola* and *P. brevis*, which are the predominant features of the gut microbiota of ruminants (22). These *Prevotella* SGBs could be a particular feature in carbohydrate utilization for hindgut-fermenting herbivores. The SGBs that were annotated to the family *Muribaculaceae* were the second most abundant taxa, accounting for a 6.81% relative abundance. *Muribaculaceae* SGBs harbored 44.17 ± 15.99 GHs, on average, as well as 47 versatile GHs families, with GH13, GH2, and GH3 being the most abundant. The family *Muribaculaceae*, which belongs to the *Bacteroidia* class, is historically dominant in rodents (24). Consistent with the previous results reported by Lagkouvardos (25), the species of *Muribaculaceae* isolated from mice also displayed a high occurrence of GH13, which implicated alpha-glucan degradation. Although the SGBs of *Treponema* that belong to *Spirochaetota* were characterized by the highest relative abundance of 9.48%, they carried a smaller mean number of GHs (28.00 ± 8.68), and they were involved in 31 different GHs families. *Treponema* SGBs also mainly contain the GH13 family, and this is highly similar to what was observed regarding the glycogen-debranching enzyme GlgX (Table S9). Collectively, the CAZymes of plateau

**FIG 5** Legend (Continued)
metabolite. An edge was drawn between the SGBs and the metabolites if they showed a conditional probability ≥4.5. (B) Proposed mechanism for the tryptophan synthesis and metabolism pathway. Gene organization of the putative tryptophan synthesis *trp*-like gene operon and a set of enzymes containing 2-hydroxyacyl-CoA dehydratases that catalyze the conversion of tryptophan to IA.

pika gut microbiota, especially the GHs family, have high diversity and metabolic versatility, and SGBs belonging to the genus *Prevotella* and the family *Muribaculaceae* may play central roles.

Indoleacrylic acid, which is a fermentation product of tryptophan metabolism, has been reported to stimulate IL-10 production, inhibit the production of TNF and IL-6, and upregulate the expression of antioxidant pathways by activating NRF2. Therefore, IA promotes host anti-inflammatory responses and has therapeutic benefits for inflammatory bowel disease (IBD) (26). As the metabolite that is most correlated with pika gut microbiota, tryptophan is able to reduce oxygen consumption and maintain body temperature in cold environments (27).

Among the affiliated taxa of the core microbiota in the intestines of plateau pika, there are many probiotic microbes. *Akkermansia muciniphila* has a beneficial effect on diabetes, obesity, and colorectal cancer (CRC) (28). Recent research indicates that *Treponema* spp. are prevalent in individuals living in traditional rural societies and may have benefits for chronic diseases (29). *Muribaculaceae* is a family of uncultured bacteria that primarily inhabit mice and other rodents, and they are possibly linked to the longevity of their hosts (25). All of these suggest that the intestinal bacteria of plateau pika potentially have probiotic properties. There are also pathogenic species that are related to the core microbiota. *Bilophila wadsworthia* is associated with abscesses, appendicitis, and colitis. It can utilize taurine as a sulfite source to produce hydrogen sulfide ($H_2S$), which is a crucial factor in developing inflammatory bowel diseases (30, 31). We detected a similar taurine desulfonation pathway in *Desulfovibrionaceae bacterium HAN036* to that of *B. wadsworthia* (Fig. S8). Hence, the gut microbiota may have "double faces", and we will approach wild plateau pika with caution.

Although this metagenomic-driven investigation has expanded our views on the diversity of microbial life, it is crucial to isolate and culture species from these uncultured lineages to test genome-based predictions of their metabolic functions and physiology as well as to properly elucidate their ecological roles (32). For example, some novel enzymatic reactions and pathways can only be discovered through the experimental testing of live bacteria and cannot be detected by genomic sequencing alone (33). The species-level core gut microbiota of the plateau pika consists entirely of uncultured bacteria that are affiliated with anaerobic lineages. To enrich these hard-to-cultivate bacteria, we must design selective nutrient media and set up proper physicochemical conditions (for example, temperature, pH, and gas-phase composition). Through a function capacity analysis, we realized that the core microbiota of plateau pika could efficiently digest the plant polysaccharide and could live in a unique metabolic microenvironment (for example, an environment with high levels of oleamide, tryptophan, and hypoxanthine). Based on current knowledge, low-nutrient media with plant polysaccharide as the sole carbon source and complemented with metabolites as growth factors would help the isolation of the uncultured core microbiota of plateau pika. In addition, some innovative techniques (for example, reverse genomics [34], live-FISH [35], and SlipChip [36]) are also worth trying.

As a result of technological advancements and deeper environmental exploration, knowledge of microbial biodiversity continues to expand, based on cultivation strategies and sequencing methods. The cultivation of microorganisms is constrained by environmental conditions, which restrict our investigation of uncultured microorganisms. Via sequencing metagenomic samples and binning assembly data, we were able to reconstruct 109 species-level genome bins (SGBs) of high quality. We taxonomically assigned these uncultured bacteria with high precision by using a combination of genome-based taxonomy (GTDB-Tk), ANI, AAI, and phylogenomic analyses. 16S rRNA sequencing yielded results consistent with the core microbiota that were detected. Even though we could not obtain a complete correspondence between the SGBs and the core OPU taxa, 11 of the 13 core OPU taxa share similar phylogenetic positions to those of the SGBs. Their relative abundance values are nearly equivalent (Fig. S5). In addition, this facilitated the notion that DNA sequencing data could serve as the

primary "type material" for uncultured microorganisms and contributed to a better understanding of the physiological capabilities, ecology, and biological evolution of the entire tree of life by enhancing the communication of microbial diversity (13, 37).

**Conclusion.** In conclusion, by using a metataxonomic approach that was based on almost full-length 16S rRNA amplicon sequences, we gave, for the first time, the taxonomic structure, at the species level, of the gut microbiota of the plateau pika, which is endemic on the Qinghai-Tibet Plateau. Remarkably, we combined this method with a metagenomic binning approach to provide an in-depth view of the core uncultured bacterial taxa that inhabit the pika intestinal tract.

## MATERIALS AND METHODS

**Sample collection.** Wild plateau pika sample collection was conducted in the summer (July) of 2016 in the Yushu Tibetan Autonomous Prefecture (Altitude: 3,942 m; N: 33°4'11," E: 96°50'39"), Qinghai Province, China, the area of which was covered with typical alpine meadow. After captured via traps, plateau pikas were immediately euthanized, and then their intestinal contents were collected and frozen at $-20°C$. A total of 105 pikas were sampled and transferred along a "cold chain" with an uninterruptible power system to our laboratory in Beijing within 24 h. All of the samples were stored at $-80°C$ prior to the metagenomic and metabolomic analyses. This research followed the spirit of science and humanity, and the ethical practices were approved by the Ethical Committee of the National Institute for Communicable Disease Control and Prevention, Chinese Center for Disease Control and Prevention (number ICDC 2016004).

**Full-length 16S rRNA-based amplicon sequencing and sequence trimming.** Total DNA was extracted from each pika intestinal content sample using a DNeasy PowerSoil DNA Isolation Kit (Qiagen, Hilden, Germany), following the manufacturer's recommendations. For the analysis of the gut microbiota, we used the universal 27F/1492R (5'-AGAGTTTGATCCTGGCTCAG-3'/5'-GNTACCTTGTTACGACTT-3') primer pairs for the amplification of the full-length V1-V9 region of the microbial 16S rRNA gene. Primers containing unique 16 bp barcodes were used to tag the PCR product of each sample, which allowed us to simultaneously sequence multiple samples. PCR was performed using *Ex Taq* DNA polymerase (TaKaRa, Dalian, China). The thermal cycler procedures were: an initial denaturation step at 95°C for 3 min, followed by 28 cycles of 95°C for 30 s, 54°C for 30 s, 72°C for 90 s, and a final extension at 72°C for 10 min. Amplicons were purified using a QIA Quick PCR Purification Kit (Qiagen, Hilden, Germany). The integrity was then determined using an Agilent 2100 Bioanalyzer (Agilent, USA) and agarose gel electrophoresis, and the purity and concentration were detected using a NanoDrop 2000 (Thermo Fisher, Shanghai, China). Library preparation and sequencing were performed according to the PacBio-RSII protocol using P6/C4 chemistry with 20 h movies at TianJin Biochip Corporation (Tianjin, China).

The raw sequencing data were initially processed based on the PacBio SMRT Link (version 6.0.0) pipeline. (1) Reads from multiple samples were demultiplexed, according to the RSII_384_Barcodes with a minimum barcode score of 26. (2) To reduce the error rate, circular consensus sequencing (CCS) reads were derived from the multiple alignments of subreads and filtered with a minimum of 5 CCS passes and a minimum predicted accuracy of 99.9%. Subsequently, a QIIME (38) pipeline was applied to filter out ambiguous bases, low-quality sequences, primers, and adapters. Sequences outside the expected size (<1,200 and >1,600 bp) were trimmed. Chimeras were detected and removed using the UCHIME algorithm implemented in USEARCH (39, 40) (version 11.0.667; option: -uchime_ref -strand plus -nonchimeras) with the RDP Gold reference database (41).

**Operational phylogenetic units (OPU) assignment.** The detailed analysis procedures were described in our previous studies with slight modification (3). In brief, all of the high-quality sequences were clustered into OTUs at an identity of 98.7% via the USEARCH pipeline (version 11.0.667). The most frequent sequence from each OTU cluster was selected as the representative sequence. Representative OTU sequences were aligned using the LTP 132 database and the SINA (version 1.2.11) alignment tool. The aligned sequences were inserted into the LTP 132 database using the parsimony tool that is implemented in the ARB (version 6.0.6) software package. Unaffiliated sequences were transferred into the nonredundant SILVA Ref 132 database, and the three closest relatives for each sequence were picked out. Then, the selected relatives from the nonredundant SILVA Ref 132 database and LTP 132 database containing OTUs were merged, and a phylogenetic tree was built utilizing the neighbor joining (NJ) algorithm with the Jukes-Cantor correction. A 30% conservational filter was employed to remove hypervariable positions.

Representative OTU sequences were grouped into OPUs (operational phylogenetic units), and these were derived from a manual inspection of the NJ tree. An OPU was defined as the smallest monophyletic clade containing OTU representatives that were affiliated with at least one reference sequence. The type sequence from the LTP database was previously selected as the OPU reference, and when this was not possible, sequences from the nonredundant SILVA Ref 132 database were used (42). Generally, one OPU was equalized to a prokaryotic species, owing to the high internal consistency. (i) When an identity value of OTU representatives with a type strain was >98.7%, it was identified as the same known species. (ii) For identity values <98.7% and >94.5% with the closest type strain, the OTUs were considered to belong to the same genus and were assigned as a potential new species. (iii) When no type strain was close enough, the sequences were clustered into families or higher taxa and were assigned as unclassified high hierarchical taxa. The most frequent sequence from each OPU cluster was selected as the representative sequence (43).

**Metagenomic sequencing.** The DNA used for metagenomic sequencing was the same material as described in the 16S rRNA-based amplicon sequencing subsection. DNA degradation and contamination were monitored on 1% agarose gels. DNA concentration was determined using a Qubit 2.0 Flurometer with a Qubit dsDNA Assay Kit (Life Technologies, CA, USA). A total amount of 1 $\mu$g DNA per sample was employed for the library preparations. Sequencing libraries were constructed using the NEBNext Ultra DNA Library Prep Kit (NEB, USA), following the manufacturer's instructions, and index codes were incorporated to attribute sequences to each sample. The libraries were purified using an AMPure XP system, analyzed for size distribution using an Agilent2100 Bioanalyzer (Agilent, USA), and quantified using real-time PCR. At last, library preparations were sequenced on an Illumina HiSeq 2000 platform (150 bp, paired-end) at Novogene Co., Ltd. (Beijing, China). Hence, in total, 16 DNA samples were sequenced (average size of 6.1 Gbp) (see Table S5 for details).

**Metagenomic assembly and contig binning.** The raw paired-end reads were quality-filtered using the KneadData pipeline (version 0.7.3; http://huttenhower.sph.harvard.edu/kneaddata), which integrates the tools Trimmomatic (44) (version 0.39; option "SLIDINGWINDOW:4:30 MINLEN:60") and Bowtie2 (version 2.3.5; option "-very-sensitive -dovetail") to do the read trimming and decontamination of the Ochotona host DNA (GCA_000292845.1_OchPri3.0). The clean reads from each of the 16 metagenomes were independently assembled with metaSPAdes (45) (version 3.13.0; default parameters), which exhibited superior accuracies among the metagenomic assemblers. Compared with the coassembly, the independent assembly could generate more and higher-quality genomes from metagenomic data sets because of the lower complexity of the individual samples. The assembled contigs were binned using three different tools, using the default settings in the MetaWRAP (46) (version 1.3.2) pipeline, including MetaBAT2, MaxBin2, and CONCOCT. The MAGs recovered by the above three algorithms were consolidated with the "Bin_refinement" module (option: completeness > 50%; contamination < 10%), and they were then reassembled using the "Reassemble_bins" module implemented in MetaWRAP (version 1.3.2). Subsequently, the reassembled MAGs were dereplicated using dRep (47) (version 3.0.1; option: -comp 50 -con 10) to construct a nonredundant MAG set.

**Organizing metagenomic assemblies into species-level genome bins (SGBs).** Based on the quality, as estimated using CheckM (version 1.0.13 [48]), the reconstructed 747 MAGs were divided into three quality levels (49): (i) near-complete, high-quality genomes (completeness $\geq$ 90%; contamination $\leq$ 5%); (ii) M=medium-quality genomes (completeness $\geq$ 70%; contamination $\leq$ 10%); and (iii) partial genomes (completeness $\geq$ 50%; contamination $\leq$ 5%). To achieve a good signal (completeness) to noise (contamination) ratio, a stricter quality control was adopted, with completeness $-5 \times$ contamination, and only MAGs that passed the control were used in further analyses (13).

To investigate the diversity of the microbial species that inhabited the plateau pika gut and to generate a comprehensive set of high-quality reference genomes, we clustered the total set of 165 strictly quality controlled, near-complete MAGs at an estimated species level (ANI $\geq$ 95%) using dRep (version 3.0.1; option: -pa 0.9 -sa 0.95 –S_algorithm fastANI -cm larger). The best MAGs were selected as representative genomes of each SGB based on the dRep quality score.

$$\text{Genome Quality} = \text{Completeness} - (5 \times \text{Contamination}) + (\text{Contamination} \times (\text{Strain Heterogeneity}/100)) + 0.5 \times (\log(\text{N50})$$

**Phylogenetic analyses of the plateau pika microbiome.** The phylogenetic analyses of the full-length 16S rRNA-based amplicon sequencing were performed using RAxML (50) (version 8.2.12) as follows. (i) The representative sequences of core OPUs, which exited $\geq$ 60% of all samples, were picked out. (ii) 297 type strain sequences, belonging to 7 classes in which core OPUs were embedded, were downloaded from the LTPs 132 SSU database. (iii) The combination of the core OPUs and the 297 references were aligned using MAFFT (51) (version 7.464; default parameters). (iv) The core OPUs phylogenetic tree was inferred with the alignment using RAxML (version 8.2.12; option: -# 100 -m GTRCAT -p 1234). (v) iTOL (https://itol.embl.de/) was employed to visualize and annotate the phylogenetic tree (52).

The phylogenetic analyses of 109 SGBs were performed using PhyloPhlAn (53). The phylogeny was built using the 400 universal PhyloPhlAn markers with the following options: "-diversity high -accurate -min_num_markers 80". The following internal steps, with their sets of parameters, were used:

diamond (54) (version v0.9.9.110) with parameters "blastx -quiet -threads 1 -outfmt 6 -more-sensitive -id 50 -max-hsps 35 -k 0" and with parameters "blastp -quiet -threads 1 -outfmt 6 -more-sensitive -id 50 -max-hsps 35 -k 0";

mafft (version v7.310, [51]) with the "-anysymbol" option;

trimal (version 1.2rev59, [55]) with the "-gappyout" option;

RAxML (version 8.1.15, [50]) with parameters "-#100 -m PROTCATLG -p 1989".

The phylogenetic trees of the core bacteria were generated from core-genome sequences using roary (56) (see Table S8 for the details of the core genes). These were inferred via RAxML methods (version 8.2.12; option: -# 100 -m PROTGAMMAAUTO -p 1234).

**Taxonomic assignment of SGBs.** The taxonomic classification of SGBs was initially performed using GTDB-Tk (57) (version 1.3.0), based on the GTDB RefSeq_95 database. Consistent with the taxonomic results of the core OPUs, none of the 109 SGBs could be clearly assigned to a named species. Furthermore, the genome-derived ANI (average nucleotide identity), AAI (average amino-acid identity), and phylogenetic analysis were then employed and integrated to obtain an accurate taxonomic classification of the SGBs. The ANI values that were calculated via the OrthoANI algorithm (58) (ChunLab) provided robust species boundaries ($\geq$95% ANI) (59) to assess whether SGBs belonged to a named bacterial

species. For more distantly related populations, the AAI values were calculated via comparem (version 0.1.1; option: aai_wf) and applied to estimate higher taxa circumscribes: 45 to 65%, 65 to 95%, 95% to 100% for the (same) family, genus, and species, respectively (10). Note that overlap exists among different ranks (species, genus, family), as the intertaxon divergence is significantly higher than the intrataxon diversity. Hence, the phylogenetic affiliations of the SGBs were examined to assist with the adjustment of the taxonomic ranks (i.e., if one SGB had $\leq$ 65% AAI from all of the named species while it embedded in the clade of a known genus, we assigned this organism to a novel species rather than to a novel genus).

**Metabolites extraction.** Pika intestinal content samples (100 $\mu$L) were individually grounded with liquid nitrogen. The homogenate was resuspended with prechilled 80% methanol and 0.1% formic acid, and it was well vortexed. Then, the mixture was incubated on ice for 5 min and centrifuged at 15,000 $\times$ $g$ and 4°C for 20 min. Some of the supernatant was diluted to a final concentration containing 53% methanol. The samples were subsequently transferred to a fresh Eppendorf tube and centrifuged at 15,000 $\times$ $g$ and 4°C for 20 min. Finally, the supernatant was injected into an LC-MS/MS system.

**UHPLC-MS/MS analyses.** UHPLC-MS/MS analyses were carried out using a Vanquish UHPLC system (Thermo Fisher, Germany) that was coupled with an Orbitrap Q Exactive HF mass spectrometer (Thermo Fisher, Germany) at Novogene Co., Ltd. (Beijing, China). The samples were injected into a Hypesil Goldcolumn (100 $\times$ 2.1 mm, 1.9 $\mu$m) using a 17 min linear gradient with a flow rate of 0.2 mL/min. The elution of the positive polarity mode was performed using eluent A (0.1% FA in water) and eluent B (methanol). The elution of the negative polarity mode was performed using eluent A (5 mM ammonium acetate, pH 9.0) and eluent B (methanol). The solvent gradient was set as follows: 2% B, 1.5 min; 2 to 100% B, 12.0 min; 100% B, 14.0 min; 100 to 2% B, 14.1 min; and 2% B, 17 min. The Q Exactive HF mass spectrometer was operated in the positive/negative polarity mode. The capillary temperature was 320°C, and the sheath gas flow rate was set at 40 arb. The source voltage was 3.2 kV for both modes.

**Metabolomics data processing and metabolite identification.** Raw data were preliminary processed using the Compound Discoverer (version 3.1, Thermo Fisher) to execute peak alignment, peak picking, and quantitation for the metabolites. The key parameters were optimized as follows: 0.2 min for retention time tolerance; 5 ppm for actual mass tolerance; 30% signal intensity tolerance; signal/noise ratio, 3; and minimum intensity, 100,000. Thereafter, the peak intensities were normalized to the total spectral intensity and used to predict the molecular formula, based on additive ions, molecular ion peaks, and fragment ions. Then, the peaks were searched with the mzCloud (https://www.mzcloud.org/), mzVault, and MassList database to generate the accurate qualitative and relative quantitative results. The metabolites were annotated using the KEGG (https://www.genome.jp/kegg/pathway.html), HMDB (https://hmdb.ca/metabolites), and LIPIDMaps (http://www.lipidmaps.org/).

**CAZymes analysis.** The annotations for the carbohydrate-active enzymes (CAZymes) were adopted against the CAZy database using dbcan (60) (version 2.0.11).

**Correlation analysis.** The cooccurrences of microbes and metabolites were determined using mmvec (61) (version 1.0.6; option -p-num-testing-examples 1 -p-latent-dim 2 -p-min-feature-count 0 -p-epochs 6,000) and visualized using the *igraph* package (62) implemented in R (version 4.0.0).

**Data availability.** The trimmed 16s rRNA sequences, metagenomic data sets, and recovered high-quality SGBs are deposited in the NCBI GenBank database under one BioProject with the accession number PRJNA692773. The metabolomic data sets are deposited in figshare with the digital object identifier (DOI) .

## SUPPLEMENTAL MATERIAL

Supplemental material is available online only.
**SUPPLEMENTAL FILE 1**, XLSX file, 0.02 MB.
**SUPPLEMENTAL FILE 2**, XLSX file, 0.6 MB.
**SUPPLEMENTAL FILE 3**, XLSX file, 0.01 MB.
**SUPPLEMENTAL FILE 4**, XLSX file, 0.01 MB.
**SUPPLEMENTAL FILE 5**, XLSX file, 0.02 MB.
**SUPPLEMENTAL FILE 6**, XLSX file, 0.03 MB.
**SUPPLEMENTAL FILE 7**, XLSX file, 0.02 MB.
**SUPPLEMENTAL FILE 8**, XLSX file, 0.1 MB.
**SUPPLEMENTAL FILE 9**, XLSX file, 0.7 MB.
**SUPPLEMENTAL FILE 10**, PDF file, 6.6 MB.

## ACKNOWLEDGMENTS

We thank the TianJin Biochip Corporation (Tianjin, China) for the technical assistance with the PacBio sequencing.

J.P. analyzed the data and wrote the first draft of the manuscript. J.P., Y.X., S.L., L.X., Y.H., S.W., and X.B. collected the multiomics samples. L.N. processed and extracted the sample DNA. J.Y., D.J., and W.Z. contributed to the PacBio and Illumina sequencing. J.P. contributed to the development of the metabolomic, metagenomic, and full-length

16S rRNA pipeline. J.X. conceived the study and guided the interpretation. J.X., J.Y., and L.L. acquired the funding. All authors contributed to the final version of the manuscript.

This work was supported by grants from the National Key R&D Program of China (2019YFC1200501, 2019YFC1200505) and the Research Units of Discovery of Unknown Bacteria and Function (2018RU010).

We declare no competing interests.

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
