## [Reviewer comments · Microbiology Spectrum]

Microbiology Spectrum

Species-level taxonomic characterization of uncultured core gut microbiota of plateau pika

Ji Pu, Jing Yang, Shan Lu, Dong Jin, Xuelian Luo, Yanwen Xiong, Xiangning Bai, Wentao Zhu, Yuyuan Huang, Shusheng Wu, Lina Niu, Liyun Liu, and Jianguo Xu

Corresponding Author(s): Jianguo Xu, State Key Laboratory of Infectious Disease Prevention and Control

Review Timeline:

Submission Date:	August 31, 2022
Editorial Decision:	September 28, 2022
Revision Received:	November 29, 2022
Editorial Decision:	December 21, 2022
Revision Received:	February 12, 2023
Accepted:	February 13, 2023

Editor: Jianjun Wang

Reviewer(s): The reviewers have opted to remain anonymous.

Transaction Report:

DOI: <https://doi.org/10.1128/spectrum.03495-22>

September 28, 2022

Prof. Jianguo Xu

State Key Laboratory of Infectious Disease Prevention and Control, National Institute for Communicable Disease Control and Prevention, Chinese Center for Disease Control and Prevention
Changping, Beijing 102206
China

Re: Spectrum03495-22 (Species-level taxonomic characterization of uncultured core gut microbiota of plateau pika)

Dear Prof. Jianguo Xu:

Thank you for submitting your manuscript to Microbiology Spectrum. Your manuscript has been reviewed by two referees in this field, and both of them saw your findings as valuable but suggested further revision. When submitting the revised version of your paper, please provide (1) point-by-point responses to the issues raised by the reviewers as file type "Response to Reviewers," not in your cover letter, and (2) a PDF file that indicates the changes from the original submission (by highlighting or underlining the changes) as file type "Marked Up Manuscript - For Review Only". Please use this link to submit your revised manuscript - we strongly recommend that you submit your paper within the next 60 days or reach out to me. Detailed instructions on submitting your revised paper are below.

Link Not Available

Sincerely,

jianjun wang

Journals Department
Reviewer comments:

Reviewer #1 (Comments for the Author):

The manuscript by Ji Pu and colleagues explores the species-level uncultured core gut microbiota of plateau pika. I must admit that the manuscript was very interesting and the science behind well executed. I really enjoyed reading it, although I have some questions arising from the data presented in this manuscript:

Comments:

The presentation of Fig 1 does not provide comprehensive information about the identified 618 OPU. Given the 16S full-length sequences, I believe that three categories of 618 OPUs marked in one phylogenetic tree would be more comprehensive. The readability of SGBs phylogenetic tree in Fig 2 is very poor. The SGBs are only in a few clades, but many poorly related phylogenetic clades are fully shown, resulting that SGBs are not obvious. Collapsing the poorly related phylogenetic clades

would be helpful to highlight SGBs.

Is there descriptions of Fig 3 between Fig 2 and Fig 4 (lines 157-159)? Do the results correspond to the descriptions one by one?

Is there a relationship between the sampling time (July, 2016) with species-level core microbiota? There would be some auxiliary microbiota in so-called "core microbiota" based on the samples in each time period.

Based on the metabolic reconstruction of SGBs assigned uncultured taxonomic groups, supplying more discussions about how to isolate uncultured resources would be helpful to expatiate the importance of studying the species-level core microbiota.

Lines 92-94: it would be confusing that the sequences of OTU were picked out for phylogenetic inference with LTP and SILVA databases. The phylogenetic inference concerned with the sequence alignment; however, the main difference between LTP and SILVA databases is taxonomy (<https://www.arb-silva.de/documentation/release-132/>).

Line 130: Akkermansiaceae should be *Akkermansiaceae*. Italic face format.

Lines 143-144: The format of ">" and "<" did not consist with others.

Line 188: 3692 should be 3,692.

Line 189: 1604 should be 1,604.

Lines 189-192: the commas in parentheses were not in "Times New Roman".

Line 197: 1547 should be 1,547.

Lines 224-225: Which part of the data should this result refer to?

Line 295: 3942 should be 3,942.

Line 328: "QIIME (30)pipeline" should "QIIME (30) pipeline". Need a space.

Lines 331-332: there was no reference to USEARCH. More useful information would be found on the web (<https://www.drive5.com/usearch/manual/citation.html>).

Line 332: there was no reference to RDP Gold database (<http://rdp.cme.msu.edu/misc/citation.jsp>).

Line 375: "Table S5 for details)" should be "Table S5 for details)".

Line 380: "version 2.3.5;option" should be "version 2.3.5; option". Need a space.

Line 396: "(version 1.0.13(38))" should be "(version 1.0.13 (38))". Need a space.

Lines 424-431: The format of double quotation marks did not consist with other in the manuscript. The typeface of double quotation marks was not in "Times New Roman".

Line 469: The typeface of semicolon was not in "Times New Roman".

Line 458: 15000 should be 15,000.

Line 469: The typeface of double quotation marks was not in "Times New Roman".

Line 657: the format of citation was not consistent with others.

Reviewer #2 (Comments for the Author):

This is a good paper describing taxonomic structure of the gut microbiota of the plateau pika that endemic on Qinghai-Tibet Plateau. Remarkably, the authors combined metagenomic binning approach to provide an in-depth view of the core uncultured bacterial taxa inhabited in the pika intestinal tract. I was impressed by the amount of work the authors have put into this study, and their manuscript was carefully written up. Generally, results have been well explained and the figures are of high quality. So, all in all, I think this is a great manuscript with a very good fit for Microbiology Spectrum.

I have several minor points for clarifications that may improve the manuscript (see below).

Minor comments:

1. If possible, please provide metadata information of 105 intestinal content samples, so that readers can better understand the plateau's living environment and population representation.
2. The "tax" in line 106 should be changed to "taxa".
3. It is recommended to rewrite lines 121-134 and put the OPU information in the supplementary table.
4. It is suggested to rephrase lines 162-177. The taxa with relative abundance greater than 1% are provided in the text, and the rest taxa are presented in the supplementary table.
5. Pikas are herbivorous, consuming high levels of moss, which is exceptionally high in fibre and low in protein. Therefore, the genes related to cellulose degradation and fermentation should be numerous and valuable. It is suggested that the author focus on the genes related to cellulose degradation and fermentation in subsequent research.
6. In some places, figure legends are inadequate, such as Fig 4.
7. There have some misleading descriptions in this paper. If being polished by a native English speaker, the paper would highly benefit.

Staff Comments:

Preparing Revision Guidelines

Please return the manuscript within 60 days; if you cannot complete the modification within this time period, please contact me. If you do not wish to modify the manuscript and prefer to submit it to another journal, please notify me of your decision immediately so that the manuscript may be formally withdrawn from consideration by Microbiology Spectrum.

This is a good paper describing taxonomic structure of the gut microbiota of the plateau pika that endemic on Qinghai-Tibet Plateau. Remarkably, the authors combined metagenomic binning approach to provide an in-depth view of the core uncultured bacterial taxa inhabited in the pika intestinal tract. I was impressed by the amount of work the authors have put into this study, and their manuscript was carefully written up. Generally, results have been well explained and the figures are of high quality. So, all in all, I think this is a great manuscript with a very good fit for Microbiology Spectrum.

I have several minor points for clarifications that may improve the manuscript (see below).

Minor comments:

1. If possible, please provide metadata information of 105 intestinal content samples, so that readers can better understand the plateau's living environment and population representation.
2. The "tax" in line 106 should be changed to "taxa".
3. It is recommended to rewrite lines 121-134 and put the OPU information in the supplementary table.

4. It is suggested to rephrase lines 162-177. The taxa with relative abundance greater than 1% are provided in the text, and the rest taxa are presented in the supplementary table.
5. Pikas are herbivorous, consuming high levels of moss, which is exceptionally high in fibre and low in protein. Therefore, the genes related to cellulose degradation and fermentation should be numerous and valuable. It is suggested that the author focus on the genes related to cellulose degradation and fermentation in subsequent research.
6. In some places, figure legends are inadequate, such as Fig 4.

Dear Editor and Reviewer

We deeply appreciate the effort that you and the reviewers put into carefully reviewing our work; the manuscript (ID: Spectrum03495-22) has now been revised according to your thoughtful and helpful comments. A clean version of our revised manuscript, consisting of the main text, tables, and separate figure files, has been submitted via the Editorial Manager and along with a marked-up version. Our responses to the reviewer's comments are included at the end of this letter.

Sincerely,

Jianguo Xu

Responses to Reviewer #1

Q1: The presentation of Fig 1 does not provide comprehensive information about the identified 618 OPU. Given the 16S full-length sequences, I believe that three categories of 618 OPUs marked in one phylogenetic tree would be more comprehensive.

Response: Thank you for your suggestion. We reconstructed the phylogenetic tree using the full-length 16S rRNA sequences to provide comprehensive information. The three categories of 618 OPUs were marked in one tree. Moreover, we highlighted the relative abundance and prevalence rate of each 618 OPUs to clarify that the highly prevalent OPUs were uncultured bacteria. We have uploaded the modified Fig 1 in the revised version (Lines 725 – 732 in the revised manuscript).

Q2: The readability of SGBs phylogenetic tree in Fig 2 is very poor. The SGBs are only in a few clades, but many poorly related phylogenetic clades are fully shown, resulting that SGBs are not obvious. Collapsing the poorly related phylogenetic clades would be helpful to highlight SGBs.

Response: We agree with your comments. There were so many poorly related clades in the phylogenetic tree. To highlight the phylogenetic position of SGBs, we keep the genome sequences of 7 classes (Deltaproteobacteria, Verrucomicrobiae, Bacteroidia, Spirochaetia, Epsilonproteobacteria, Clostridia, Bacilli) where SGBs affiliated to. The modified Fig 2 was uploaded in the revised version (Lines 734 – 739 in the revised manuscript).

Q3: Is there descriptions of Fig 3 between Fig 2 and Fig 4 (lines 157-159)? Do the results correspond to the descriptions one by one?

Response: Thank you very much for pointing it out. We have revised the relevant

sections of the paper, and the descriptions of Fig 3 did exist between Fig 2 and Fig 4. We have added the reference to Fig 3 appropriately in the descriptions. The number of SGBs is greater than the core OPUs, and only a part of them represented the core microbiota. Moreover, differentiating which SGB is the same species as the core OPUs is challenging. Hence, we corrected the number of SGBs representing the same dominant core microbial lineages detected by the amplicon sequencing. Only the SGBs in the same position on the phylogenetic tree have similar relative abundance with the core OPUs counted and highlighted in Fig 3.

We made the following revision:

“20 out of 109 (18.35%) SGBs (Fig. 3) represented the same dominant core microbial lineages as previously determined by the full-length 16S rRNA gene amplicon sequencing” (Lines 168 – 170 in the revised manuscript). The modified Fig. 3 was submitted.

Q4: Is there a relationship between the sampling time (July, 2016) with species-level core microbiota? There would be some auxiliary microbiota in so-called "core microbiota" based on the samples in each time period.

Response: We agree with your comments. There were some reports about the seasons' factors influencing the microbiota of plateau pika (1). In the present research, we investigate the “core microbiota” in a large number of plateau pika only at one time point. We chose July as the sampling time because the population density of pika reached its peak during this period. We also have been making an effort to keep surveillance of the gut microbiota of plateau pika through a wide range of time. However, the sequencing data were still in curation. We hope to bring a more comprehensive study of the “core microbiota” of plateau pika in the future.

Q5: Based on the metabolic reconstruction of SGBs assigned uncultured taxonomic

groups, supplying more discussions about how to isolate uncultured resources would be helpful to expatiate the importance of studying the species-level core microbiota.

Response: Thank you for your comments. We have added the relevant content in the discussion section.

We made the following revision (Lines 270 – 287 in the revised manuscript):

“Although the metagenomic-driven investigation has expanded our views on the diversity of microbial life, it is crucial to isolate and culture species from these uncultured lineages to test genome-based predictions of their metabolic functions and physiology and properly elucidate their ecological roles. Such as, some novel enzymatic reactions and pathways could only be discovered through experimental testing of live bacteria and not be detected by genomic sequencing alone. Plateau pika's species-level core gut microbiota is all uncultured bacteria affiliated with the anaerobic lineages. To enrich these hard-to-cultivate bacteria, we must design selective nutrient media and set up proper physicochemical conditions (for example, temperature, pH, and gas-phase composition). Through the function capacity analysis, we realized that the core microbiota of plateau pika could efficiently digest the plant polysaccharide and live in a unique metabolic microenvironment (for example, with a high-level oleamide, tryptophan, and hypoxanthine). Based on the current knowledge, the low-nutrient media with plant polysaccharide as the sole carbon source, and complement with metabolites as growth factors, would help the isolation of the uncultured core microbiota of plateau pika. In addition, some innovative techniques (for example, reverse genomics, live-FISH, and SlipChip) are also worth trying.”

Q6: Lines 92-94: it would be confusing that the sequences of OTU were picked out for phylogenetic inference with LTP and SILVA databases. The phylogenetic inference concerned with the sequence alignment; however, the main difference

between LTP and SILVA databases is taxonomy (<https://www.arb-silva.de/documentation/release-132/>).

Response: Sorry for the confusion caused by OTU phylogenetic inference. Our phylogenetic inference and taxonomic assignment were mainly based on the LTP database. We selected the relative reference sequences from the SILVA databases to supplement when there is no affiliated relative of OTU sequences in the LTP database. A detailed description can be found in the “Operational Phylogenetic Units (OPU) Assignment” section (lines in 356-383) of material and methods.

To avoid confusion, we made the following revision:

“The representative sequences of each OTU were picked out for phylogenetic inference” (lines 107-108 in the revised manuscript).

Q7: Line 130: Akkermansiaceae should be *Akkermansiaceae*. Italic face format.

Response: We have changed “Akkermansiaceae” to “*Akkermansiaceae*” in italic face format (line 140 in the revised manuscript).

Q8: Lines 143-144: The format of ">" and "<" did not consist with others.

Response: We have changed the “>” and “<” to the consist format with others (lines 153-154 in the revised manuscript).

Q9: Line 188: 3692 should be 3,692.

Response: We have changed “3692” to “3,692” (line 190 in the revised manuscript).

Q10: Line 189: 1604 should be 1,604.

Response: We have changed “1604” to “1,604” (line 191 in the revised manuscript).

Q11: Lines 189-192: the commas in parentheses were not in "Times New Roman".

Response: We have changed the four commas in parentheses in “Times New Roman” format (lines 191-194 in the revised manuscript).

Q12: Line 197: 1547 should be 1,547.

Response: We have changed “1547” to “1,547” (line 199 in the revised manuscript).

Q13: Lines 224-225: Which part of the data should this result refer to?

Response: This result refers to the data of “Fig. 4 A, Fig. S5 A and Table S3”, and the information was added at the end of this sentence in parentheses (lines 227-228 in the revised manuscript).

Q14: Line 295: 3942 should be 3,942.

Response: We have changed “3942” to “3,942” (line 316 in the revised manuscript).

Q15: Line 328: “QIIME (30)pipeline” should “QIIME (30) pipeline”. Need a space.

Response: We have changed “QIIME (30)pipeline” to “QIIME (35) pipeline” with a space (line 350 in the revised manuscript).

Q16: Lines 331-332: there was no reference to USEARCH. More useful information would be found on the web (<https://www.drive5.com/usearch/manual/citation.html>).

Response: Thank you for your comments and suggestions. We have added the reference to USEARCH (line 354 in the revised manuscript).

USEARCH: Edgar RC. 2010. Search and clustering orders of magnitude faster than BLAST. *Bioinformatics* 26:2460-2461.

Q17: Line 332: there was no reference to RDP Gold database

(<http://rdp.cme.msu.edu/misc/citation.jsp>).

Response: Thank you for your comments and suggestions. We have added the reference to RDP Gold database in the revised manuscript (line 355 in the revised manuscript).

The reference to RDP: Cole JR, Wang Q, Fish JA, Chai B, McGarrell DM, Sun Y, Brown CT, Porras-Alfaro A, Kuske CR, Tiedje JM. 2013. Ribosomal Database Project: data and tools for high throughput rRNA analysis. *Nucleic Acids Research* 42:633-642.

Q17: Line 375: "Table S5 for details)" should be "Table S5 for details).".

Response: We have changed “Table S5 for details)” to “Table S5 for details).” (line 397 in the revised manuscript).

Q18: Line 380: "version 2.3.5;option" should be "version 2.3.5; option". Need a space.

Response: We have changed “version 2.3.5;option” to “version 2.3.5; option” with a space (line 402 in the revised manuscript).

Q19: Line 396: "(version 1.0.13(38))" should be "(version 1.0.13 (38))". Need a space.

Response: We have changed “(version 1.0.13(38))” to “(version 1.0.13 (45))” with a space. (line 418 in the revised manuscript).

Q20: Lines 424-431: The format of double quotation marks did not consist with other in the manuscript. The typeface of double quotation marks was not in "Times New Roman".

Response: Sorry for the wrong typeface. We have corrected the double quotation

marks in “Times New Roman” typeface (lines 449-454 in the revised manuscript).

Q21: Line 469: The typeface of semicolon was not in "Times New Roman".

Response: We have changed the typeface of the semicolon to “Times New Roman” (lines 492-493 in the revised manuscript).

Q22: Line 458: 15000 should be 15,000.

Response: We have changed “15000” to “15,000” (line 479 in the revised manuscript).

Q23: Line 469: The typeface of double quotation marks was not in "Times New Roman".

Response: We have changed the typeface of the semicolon to “Times New Roman” (lines 480-481 in the revised manuscript).

Q24: Line 657: the format of citation was not consistent with others.

Response: We have corrected the citation format to make it consistent with others (line 695 in the revised manuscript).

Responses to Reviewer #2

Q1: If possible, please provide metadata information of 105 intestinal content samples, so that readers can better understand the plateau's living environment and population representation.

Response: We agree with your comments. We have added the metadata information of 105 intestinal content samples in supplementary table 1.

Q2: The "tax" in line 106 should be changed to "taxa".

Response: Thank you very much for pointing it out. We have changed the "tax" to "taxa" (line 119 in the revised manuscript).

Q3: It is recommended to rewrite lines 121-134 and put the OPU information in the supplementary table.

Response: We have rewritten lines 121-134 and put the OPU information in the supplementary table 3 in the revised manuscript (lines 134-143 in the revised manuscript).

Q4: It is suggested to rephrase lines 162-177. The taxa with relative abundance greater than 1% are provided in the text, and the rest taxa are presented in the supplementary table.

Response: We have rewritten lines 162-177, and the taxa with relative abundance lower than 1% were moved to supplementary table 6 in the revised manuscript (lines 172-179 in the revised manuscript).

Q5: Pikas are herbivorous, consuming high levels of moss, which is exceptionally high in fibre and low in protein. Therefore, the genes related to cellulose degradation

and fermentation should be numerous and valuable. It is suggested that the author focus on the genes related to cellulose degradation and fermentation in subsequent research.

Response: Thank you for your suggestion. We have added the discussion of how to use the cellulose degradation ability to isolate the core microbiota of plateau pika, and we will carry out relevant research about cellulose degradation and fermentation in the future (lines 270-287 in the revised manuscript).

Q6: In some places, figure legends are inadequate, such as Fig 4.

Response: Thank you for your suggestions. We have modified the figure legends for Fig 3 and Fig 4 (line 741-752 in the revised manuscript).

Q7: There have some misleading descriptions in this paper. If being polished by a native English speaker, the paper would highly benefit.

Response: Thank you for your comments. We have polished the revised manuscript by a native English speaker.

Reference:

1. Chao F, Liangzhi Z, Haibo F, Chuanfa L, Wenjing L, He Z, Xianjiang T, Qi C, Wenjuan S, Yanming Z. 2021. Seasonality of abundant and rare taxa in gut microbiota of plateau pikas. *ACTA THERIOLOGICA SINICA* 41:617.

December 21, 2022

Prof. Jianguo Xu
State Key Laboratory of Infectious Disease Prevention and Control
Changping, Beijing 102206
China

Re: Spectrum03495-22R1 (Species-level taxonomic characterization of uncultured core gut microbiota of plateau pika)

Dear Prof. Jianguo Xu:

Your manuscript has been seen again by the same reviewers, and you could see there are still room in improving the writing and discussion per suggested. I thus expect another round of revision is needed. Please follow carefully these comments in improving your manuscript.

Link Not Available

Sincerely,

jianjun wang

Journals Department
Reviewer comments:

Reviewer #1 (Comments for the Author):

- 1、 Line 162 remove brackets ()
- 2、 Line 293-294 "We taxonomic " There is something wrong with the expression of this sentence.
- 3、 In Lines 110-115, those percent of relative abundance is accurate to one decimal place. But in Lines 128-145, the percent of relative abundance is accurate to two decimal place. It is recommended to unify them. Maybe there are also the same problems in other part, check it.
- 4、 Line 153-154 "> 50% genome completeness, < 10% contamination and completeness - 5 * contamination {greater than or equal to} 50 " Is there a basis or reference for this standard?

5、 It is suggested that more discussion should be supplied around the Carbohydrate active enzymes of each SGB, and combining the relative abundance of different taxa, provide more information about the contribution of each taxon to the polysaccharide degradation.

Staff Comments:

Preparing Revision Guidelines

Please return the manuscript within 60 days; if you cannot complete the modification within this time period, please contact me. If you do not wish to modify the manuscript and prefer to submit it to another journal, please notify me of your decision immediately so that the manuscript may be formally withdrawn from consideration by Microbiology Spectrum.

- 1、 Line 162 remove brackets ()
- 2、 Line 293-294 “We taxonomic...” There is something wrong with the expression of this sentence.
- 3、 In Lines 110-115, those percent of relative abundance is accurate to one decimal place. But in Lines 128-145, the percent of relative abundance is accurate to two decimal place. It is recommended to unify them. Maybe there are also the same problems in other part, check it.
- 4、 Line 153-154 “> 50% genome completeness, < 10% contamination and completeness – 5 * contamination \geq 50 “ Is there a basis or reference for this standard?
- 5、 It is suggested that more discussion should be supplied around the Carbohydrate active enzymes of each SGB, and combining the relative abundance of different taxa, provide more information about the contribution of each taxon to the polysaccharide degradation.

- 1、 Line 162 remove brackets ()
- 2、 Line 293-294 “We taxonomic...” There is something wrong with the expression of this sentence.
- 3、 In Lines 110-115, those percent of relative abundance is accurate to one decimal place. But in Lines 128-145, the percent of relative abundance is accurate to two decimal place. It is recommended to unify them. Maybe there are also the same problems in other part, check it.
- 4、 Line 153-154 “> 50% genome completeness, < 10% contamination and completeness – 5 * contamination \geq 50 “ Is there a basis or reference for this standard?
- 5、 It is suggested that more discussion should be supplied around the Carbohydrate active enzymes of each SGB, and combining the relative abundance of different taxa, provide more information about the contribution of each taxon to the polysaccharide degradation.

Dear Editor and Reviewer

We deeply appreciate the effort that you and the reviewers put into carefully reviewing our work; the manuscript (ID: Spectrum03495-22) has now been revised according to your thoughtful and helpful comments. A clean version of our revised manuscript, consisting of the main text, tables, and separate figure files, has been submitted via the Editorial Manager and along with a marked-up version. Our second time responses to the reviewer's comments are included at the end of this letter.

Sincerely,

Jianguo Xu

Responses to Reviewer #1

Q1: Line 162 remove brackets ()

Response: Thank you very much for pointing it out. We have removed the brackets (Line 160 in the revised manuscript).

Q2: Line 293-294 "We taxonomic...." There is something wrong with the expression of this sentence.

Response: Thank you very much for the suggestion. We have rewritten the sentence to "We taxonomically assigned these uncultured bacteria with high precision using a combination of genome-based taxonomy (GTDB-Tk), ANI, AAI, and phylogenomic analyses." (Lines 317-320 in the revised manuscript).

Q3: In Lines 110-115, those percent of relative abundance is accurate to one decimal place. But in Lines 128-145, the percent of relative abundance is accurate to two decimal places. It is recommended to unify them. Maybe there are also the same problems in other part, check it.

Response: Thank you for the recommendation. We have carefully checked the Lines 110-115 and other part in the manuscript, and we have unified the accuracy

of relative abundance to two decimal places (Lines 101-119 in the revised manuscript).

Q4: Line 153-154 "> 50% genome completeness, < 10% contamination and completeness - 5 * contamination {greater than or equal to} 50 " Is there a basis or reference for this standard?

Response: We have attached the reference for this standard in the "RESULTS" section and the "MATERIALS AND METHODS" section (Lines 150-153 and 445-448 in the revised manuscript).

Q5: It is suggested that more discussion should be supplied around the Carbohydrate active enzymes of each SGB, and combining the relative abundance of different taxa, provide more information about the contribution of each taxon to the polysaccharide degradation.

Response: Thank you for the suggestion. We have added the relevant content in the discussion section.

We made the following revision (Lines 244-271 in the revised manuscript):
"SGBs that affiliated to the class *Bacteroidia* harbored the highest number of CAZymes (70.00 ± 29.65 on average), followed by *Clostridia* (48.98 ± 25.43), *Verrucomicrobiae* (38.00 ± 0.00) and *Spirochaetota* (34.83 ± 9.28) (Fig. S7). Among the class *Bacteroidia*, four SGBs annotated to potential new species in *Prevotella* encountered high-level GHs (76.75 ± 25.59 on average) and versatile 47 multiple

GHs families with GH43, GH2, and GH28 as the most abundant. Lineages of the *Prevotella* genus contained common member species of the human or non-human gut microbiota and have been profoundly implicated in health and disease (22). *Prevotella copri* is among the essential members prevalent in over 40% of the human population and is associated with high-fiber diets (5, 23). While in the plateau pika, the SGBs of *Prevotella* accounted for 2.23% relative abundance and were affiliated with *P. ruminicola* and *P. brevis*, which are the predominant features of ruminants' gut microbiota (22). These *Prevotella* SGBs could be a particular feature in carbohydrate utilization for hindgut-fermenting herbivores. SGBs annotated to the family of *Muribaculaceae* are the second abundant taxa that accounted for 6.81% relative abundance. *Muribaculaceae* SGBs harbored 44.17 ± 15.99 GHs on average and versatile 47 multiple GHs families with GH13, GH2, and GH3 as the most abundant. The family *Muribaculaceae* belonging to the *Bacteroidia* class is historically dominant in rodents (24). Consistently with the previous study proposed by Lagkouvardos (25), the species of *Muribaculaceae* isolated from mouse also observed a high occurrence of GH13 that implicated alpha-glucan degradation. Although the SGBs of *Treponema* belonging to the *Spirochaetota* were characterized by the highest relative abundance of 9.48%, they carried less mean GHs number with 28.00 ± 8.68 and were involved in 31 different GHs families. *Treponema* SGBs also mainly contain the GH13 family highly similar to the glycogen-debranching enzyme GlgX (Table S9). Collectively, CAZymes of plateau pika gut microbiota, especially the GHs family, have high diversity and metabolic versatility, and SGBs belonging to the genus *Prevotella* and the family *Muribaculaceae* may play central roles.”

February 13, 2023

Prof. Jianguo Xu
State Key Laboratory of Infectious Disease Prevention and Control
Changping, Beijing 102206
China

Re: Spectrum03495-22R2 (Species-level taxonomic characterization of uncultured core gut microbiota of plateau pika)

Dear Prof. Jianguo Xu:

Your manuscript has been accepted, and I am forwarding it to the ASM Journals Department for publication. You will be notified when your proofs are ready to be viewed.

Sincerely,

Jianjun Wang
Editor, Microbiology Spectrum
